# Antioxidant Enzymatic Activities and Growth Response of Quinoa (*Chenopodium quinoa* Willd) to Exogenous Selenium Application

**DOI:** 10.3390/plants10040719

**Published:** 2021-04-07

**Authors:** Ahlam Khalofah, Hussein Migdadi, Ehab El-Harty

**Affiliations:** 1Biology Department, Faculty of Science, King Khalid University, Abha 61413, Saudi Arabia; aalshayeed@kku.edu.sa; 2Research Center for Advanced Materials Science (RCAMS), King Khalid University, Abha 61413, Saudi Arabia; 3Department of Plant Production, King Saud University, College of Food and Agriculture Sciences, Riyadh 11461, Saudi Arabia; ehabelharty@gmail.com; 4National Agricultural Research Center, Baqa 19381, Jordan

**Keywords:** selenium, quinoa, antioxidant, pigments, proline, sugars

## Abstract

Selenium is a trace element essential to many organisms, including higher plants. At low concentrations, it enhances growth and development; however, it is toxic at high concentrations. The development of crops with proper levels of selenium will be worth for both nutrition and Se-based therapeutics. This study aimed to investigate the morphological, physiological, and biochemical responses of the quinoa plant to 0, 2.5, 5, 10, and 20 mg/L of Na_2_SeO_3_·5H_2_O. Selenium at low concentrations (2.5 and 5 mg/L), quinoa plant showed a significant increase of growth parameters, relative water content, photosynthetic pigments, proline, total soluble sugars, and antioxidant enzymes activities as (superoxide dismutase (SOD), catalase (CAT), peroxidase (POD, ascorbate peroxidase (APX), and glutathione reductase (GR)), and contents of malondialdehyde (MDA) and H_2_O_2_ were reduced. However, high concentrations (10 and 20) mg/L caused a decrease in plant growth parameters, relative water content, and photosynthetic pigments. In contrast, excess selenium increased the oxidative stress monitored by hydrogen peroxide and lipid peroxidation levels. The enzymatic antioxidant system responded to the selenium supply significantly increased. Osmolytes compounds, such as total sugars and proline, increased in selenium-treated plants. The increase in these osmolytes compounds may show a defense mechanism for the osmotic readjustment of quinoa plants to mitigate the toxicity caused by selenium. This study shows the morphological and physiological responses that must be considered for success in the sustainable cultivation of quinoa plants in environments containing excess selenium.

## 1. Introduction

Selenium is an essential component of human and animal cells, but it is not regarded as an essential nutrient for higher plants. However, it enhances plant growth and productivity under environmental stresses [1]. It was dealt with a toxic element until 1957 [2]. Thus, selenium’s significant impact was identified in humans, animals, and to a slighter extent in plants as essential for plant growth [3]. Soils are considered the primary source of selenium for plants, and it exists in various forms; elemental selenium, selenates, selenides, and organic selenium compounds. Soil type, climate, organic matter, and rainfall play a significant role in varied selenium content [4]. Forest soils efficiently retain selenium and then incorporate it into low-molecular-weight fractions of humic substance [5]. The availability of selenium to plant is decreased by low pH, high concentrations of sulfur and phosphorus, and soil of the world’s driest regions [6].

Cultivated plants are an essential source of selenium for humans and livestock. Being chemically analogous to sulfur, selenium is absorbed by all plants by sulfate transporters and is sequestered as selenite and selenite [7]. The world food organization considers selenium fundamental in the diet and importance of inclusion selenium in plant products, such as wheat, barley, rice, and potato [8]. Hence, several attempts were made to increase selenium content in plants, and changes in the enzymes associated with Sulphur metabolism have mainly been used to raise selenium levels in plants [9]. Selenium at low doses plays an essential role in neutralizing and protect the plants from abiotic stresses, such as cold, drought, intense light, heavy metals, water salinity, UV-b, and high temperatures [10]. Under stress conditions, reactive oxygen species are produced in plants that disrupt cell membranes, proteins, and other cell components. Selenium plays an essential role in anti-oxidation in biological organisms by stimulating many enzymes’ activity, scavenging hydrogen peroxides [7]. At a proper dose, the application of selenium is found to enhance grain yield, nutritional value, and essential elements [11]. In contrast, selenium at high concentrations harms plant growth and causes chlorosis and leaves’ death because of oxidative stress. Selenium can nonspecifically replace S in S compounds, including proteins (i.e., damaged proteins, targeted for degradation) were found in Se-treated plants [12].

Recently, *Chenopodium quinoa* Willd (quinoa) has attracted much attention because it is rich in bioactive compounds, nutritional satisfaction, and health and wellness benefits [13]. This species belongs to the *Chenopodiaceae* family and has been cultivated for centuries in the Andean countries of Peru and Bolivia [14]; cultivation has spread to several countries, such as Australia, Canada, and China, England, and others [15]. Quinoa reported standing resistance to abiotic stresses since these plants have a tremendous genetic variability that allows their adaptation and growth in the most adverse environmental conditions [15]. Quinoa is known as pseudo-cereal because, although not belonging to the *Gramineae* family, it produces seeds that can be milled into flour and used as a cereal crop [14]. Despite not being as widespread as wheat or rye, the interest in its consumption has flourished because of its attractive nutritional content. Quinoa seeds reveal an entire absence of gluten, high levels of fatty acids, vitamins, minerals, dietary fibers, and proteins with more amino acids [16]. Besides good nutritional composition, these seeds have been found to contain a large variety of bioactive compounds, such as carotenoids, vitamin C, and phenolic compounds, which are apparent in many studies as protective against a variety of diseases, especially cancer, allergy, inflammatory diseases, and may reduce the risk of cardiovascular diseases, considering quinoa seeds a functional food [17]. There are several forms of quinoa consumption; namely, the seeds can be fermented to make beer; the germinated seedlings (quinoa sprouts) can be incorporated in salads, and the whole plant can a rich nutritional source to feed livestock [18]. Since these seeds can be well milled in flour, they can be used for the same purpose as cereals, such as pastry and bakery products [18].

This study aimed to investigate the effect of selenium on morpho-physiological parameters and the antioxidant behavior of quinoa plants. Our data might highlight the changes in some metabolic processes in quinoa supplemented with Se, which might be helpful for agronomists interested in selenium fertilization. Research on selenium application on quinoa’s physiological processes is essential to understand tolerance and phytotoxicity mechanisms better.

## 2. Results

### 2.1. Plant Growth Parameters

The effects of different selenium (Se) treatments on plant growth parameters in quinoa after 30 days of treatments are presented in Table 1. All plant growth traits were significantly stimulated when treated with concentrations (2.5 and 5) mg/L of selenium compared to the control plants. In contrast, we observed a reduction in all growth parameters with concentrations of 10 and 20 Na_2_SeO_3_ mg/L (Appendix A).

### 2.2. Photosynthetic Pigment and Relative Water Contents

Selenium treatments 2.5 and 5 mg/L significantly increased the chlorophyll-a, b, total chlorophyll, and carotenoids. However, chlorophyll-a and carotenoids showed non-significant differences from control treatment at 2.5 mg/L. Compared to the control plant, the percent increase in chlorophyll-a, b, total chlorophyll, and carotenoids was 25.7, 118.5, 45.8, and 127.0%, respectively, under five mg/L selenium. While, at the higher concentrations of selenium 20 mg/L, the percent of the reduction in photosynthetic pigment contents was 35.3, 61.6, 38.7, and 25.2, respectively, for chlorophyll-a, b, total chlorophyll, and carotenoids (Appendix A).

Selenium’s application affects relative water content, chlorophyll-a, chlorophyll-b, total chlorophyll, and carotenoid contents in quinoa leaves (Table 2). The relative water content (RWC) of quinoa plants increased under treatments 2.5 and 5 mg/L of selenium, where the increase was significant at five mg/L concentration. At high levels, the reduction in RWC at the ten mg/L has significantly differed from that, at the 20 mg/L. The percent increase at five mg/L was 1.2%; however, the reduction percentage was 3.93% at 20 mg/L (Appendix A).

### 2.3. Osmolytes Contents and Oxidative Damage (ROS)

Proline and soluble sugars act as an osmo-protectant against osmotic disturbance in the plant system. Table 3 shows the proline responses and total soluble sugars contents under different concentrations of selenium. Proline and total sugar contents were significantly increased with increasing concentrations of selenium compared with control plants. The percent increase reached 102.6 in proline and 51.8 in total soluble sugars at the 20 mg/L of Na_2_SeO_3_ (Appendix A).

As induced oxidative damage in quinoa plants was measured in terms of malondialdehyde (MDA) and hydrogen peroxide (H_2_O_2_) content. A significant decrease in the level of MDA and H_2_O_2_ was observed in quinoa plants treated with selenium concentrations of 2.5 and 5 mg/L compared to control plants (Table 3). The MDA percentage decreased from 15.3 at 5 to 21.8 at 2.5 (mg/L) of Na_2_SeO_3_; however, the percent increase reached 20.3 at the higher concentration (20 mg/L) Na_2_SeO_3_. Regarding the reduction in H_2_O_2_ content, a higher percentage of 37.2% was reported at 2.5 (mg/L) of Na_2_SeO_3_. However, at 20 mg/L of Na_2_SeO_3_, the percentage increase was reached 91.1% (Appendix A).

### 2.4. Antioxidant Enzymes Activities

Table 4 illustrates that treatment quinoa plants by selenium resulted in a significant increase in the activity of superoxide dismutase (SOD), catalase (CAT), peroxidase (POD), ascorbate peroxidase (APX), and glutathione reductase (GR). The maximum activity of SOD, CAT, POD, APX, and GR by 9.0, 13.2, 20.7, 22.3, and 23.2% were recorded in the plants treated with 20 mg/L of selenium comparative with control plants (Appendix A).

## 3. Discussion

### 3.1. Plant Growth and Relative Water Content

Animals, humans, and microorganisms need selenium as an essential micronutrient [19]. Its concentration level determines this element’s biological role, and it is often labeled as a double-edged sword. For healthy growth and development, trace concentrations are necessary. It can maintain homeostatic functions at moderate concentrations, and it would be toxic at high concentrations [20,21].

Our results showed that, at a low concentration of selenium, quinoa plant growth was enhanced. Selenium at high concentrations (10 and 20) mg/L caused decreasing plant growth compared with control plants. These results coincide with that of Hartikainen et al. [22], who suggested a dual role of selenium in plants, where, at low concentrations, it can act as an antioxidant and promote plant growth, whereas, at high concentrations, it is a pro-oxidant causing metabolic disturbances and drastic yield losses in ryegrass. Hawrylak-Nowak et al. [23] reported in cucumber plants that increasing selenite concentrations increase root activity, impaired photosynthetic pigment accumulation, and chlorophyll fluorescence parameters, suggesting the upregulation of mitochondrial dehydrogenases activity. Sun et al. [24] stated that selenium at low concentrations improved the meiosis of cell meristems of plant roots, resulting in increased root length. Our result was also in agreement with many studies that promoted shoot length at low concentration [20,25] and increased leaf number by delaying leaf abscission [20]. However, at high concentrations, selenium can be toxic. Hawrylak-Nowak et al. [23] considered the reduction in the concentration of the photosynthetic pigment is the bioindicator of trace elements phytotoxicity, where the decrease in the chlorophyll concentrations is a more sensitive indicator of selenium phytotoxicity in cucumber than a reduction in plant growth, and chlorophyll b was more sensitive to the selenium stress than chlorophyll ‘a’.

Selenium increased chlorophyll content, photosynthesis, carbon fixation, and stimulated cell division [24,26], explaining why the number of quinoa leaves in the present work. In rice, an adverse effect of 1.5 mM sodium selenite on photosynthetic pigments decreased plant biomass, caused damage by excess selenium on biochemical processes, such as chlorophyll and protein biosynthesis [27]. These results were in agreement with References [21,28], who reported that high selenium concentration in the leaves could promote chlorophyll degradation and enhance oxidative stress. Kuznetsov et al. [29] suggested that in amino acids like cysteine and methionine, the replacement of sulfur atoms by selenium is behind the selenium toxicity selenium-sensitive plants. Van Hoewyk [12] reported a reduction in biomass, photosynthetic efficiency, chlorosis, and plant death. In this study, carotenoids contents were enhanced by 127% at the low concentration (5%) while the content reduced by 25.2% at the higher selenium concentration (20 mg/L). The importance of carotenoids in the cell’s protection wall from ROS activity is reported. Selenium-treated plants showed a significant decrease in carotenoids concentration, resulting in the visual symptoms of toxicity characterized as leaf chlorosis and necrosis [27] and chlorophyll oxidation resulting in low photosynthetic efficiency [30].

Selenium increased proline and total sugars concentration by 102.6% and 51.8%, respectively, at the high selenium concentration treatment (20 mg/L). These results agree with Reference [27], who reported increases in sucrose and total sugars concentration with increasing selenium concentration. Djanaguiraman et al. [31] found that proline content increased 50% in selenium sprayed plants over its control plants. The stabilization of proteins, regulation of cytosolic pH, and regulation of the nicotinamide adenine dinucleotide (NAD/NADH) ratio by scavenger of ROS could be the role of proline under oxidative stress [32]. Silva et al. [28] stated that selenium enhanced carbohydrate metabolism and mitigated oxidative stress in cowpea plants. nicotinamide adenine dinucleotide

The growth performance of the plant is mainly correlated with its water status. At low concentrations of selenium, RWC significantly increased; however, it is decreased in quinoa plants at high concentrations. The decisive role of selenium in increasing RWC has been reported in *Trifolium repens* L. [33] in barley [34] and pumpkin [35]. The effect of selenium in increasing membrane integrity [22] or decreasing photo-oxidation [36] is reported. Selenium can regulate the water status of plants and increase water uptake and tissue hydration. It may contribute to more efficient water uptake by roots or reduce transpiration intensity and reduce water loss in tissues [29]. However, at high selenium concentration, the decrease in RWC could show selenium stress, causing stomata closure and reducing atmospheric carbon fixing activation [37] or reducing the cell wall’s elasticity water content the plant [38].

### 3.2. Photosynthetic Pigments Contents

In this study, we found that application of selenium at low concentrations significantly increased chlorophyll ‘a’, b, total chlorophyll, and carotenoids, which consistent with the findings of Reference [39] in garlic, as well as Reference [34] in barley, in potato, and lettuce [40,41], and increased carotenoids of spinach occurred with selenium supplementation [20]. The decisive role of selenium has been reported in protecting photosynthetic apparatus [42], altering chlorophyll biosynthetic pathway [43], or to the efficient scavenging reactive oxygen species (ROS) because of selenium addition [44].

However, the high selenium concentrations caused a decrease in photosynthetic pigments reported in this study also agreed with Reference [45] in maize plants. The toxic physiological symptoms of stress caused by selenium in plants include reduced biomass, photosynthetic efficiency, chlorosis, and plant death [12]; therefore, the solution’s concentration at the foliar application selenium should be chosen with care. A high concentration of selenium’s adverse effects is reported to restrain the chlorophyll biosynthesis enzyme and promote chlorophyll degradation and enhance oxidative stress [25,28,30,44]. It alters the protein biosynthesis, interferes with the electron transport chain, and impairs the redox state of enzymes in chloroplasts, decreasing photosynthesis activity [12] and higher biomass plants [28]. Its case also an increase in MDA levels as a product of lipid peroxidation [44]. It was also reported that the reduction in photosynthetic pigments might be because of reduced availability of essential nutrients to metabolically active leaf tissues, resulting in leaf chlorosis, and an excess of metals has deleterious effects on the content and functionality of the photosynthetic pigments [45].

### 3.3. Osmolytes Contents

Proline accumulation is an adaptive strategy of plants to the stressful environment, which maintains the osmotic balance, scavenges excess free radicals, stabilizes cell membrane structure and function [46], regulates cellular redox potential [47] sustains photosystem II (PSII) electron transport [48]. In the present study, proline content increased with increasing concentrations of selenium. Selenium has shown that they improve photosynthetic efficiency and proline accumulation under salt stress in *Brassica juncea* and *Cucumis sativus* [49] under chilling stress in *Cucumis sativus* [50]. Wheat plants subjected to drought stress [51] or cold stress [52] showed that selenium supplementation increased proline content. Matysik et al. [32] stated that stress-inducible proline accumulation in sugarcane plants under selenium stress acts as a component of an anti-oxidative defense system rather than as an osmotic adjustment mediator. It is role under oxidative stress, including stabilizing protein complexes, regulating cytosolic pH, regulating NAD/NADH ratio, or acting as a scavenger of oxygen-free radicals.

Our result shows that the total soluble sugars significantly increased with increasing selenium concentrations, coinciding with the results of References [43,53] in different plant species. The accumulation of soluble sugars by selenium application helped wheat plants to maintain water relations, showing that selenium contributes to starch decomposition because of amylase activity under drought stress conditions [54]. Bolouri-Moghaddam et al. [55] stated that soluble sugars might involve in the oxidative pentose phosphate pathway (PPP), mitigating ROS formation in plant cells. In mung bean (*Phaseolus aureus* Roxb.) [26] observed that exposure to selenium showed increased activity of amylases and invertases. Sugar metabolism may be affected by selenium to mitigate oxidative stress and enhance carbohydrate metabolic enzymes [28]. Excess sugar production can cause increased cytosolic H_2_O_2_ concentrations [56], while Sugars’ availability also determines the reduction of energy production that contributes to eliminating ROS in chloroplasts H_2_O_2_ [55,56].

### 3.4. Oxidative Damage (ROS) and Antioxidant Enzymes Activities

In this study, we detected high levels of MDA and H_2_O_2_ with high selenium concentrations treatments (10 and 20 mg/L), showing an increase in the level of oxidative stress and membrane degradation. However, the reduction was significant at the lower concentration (2.5 and 5 mg/L), resulting in low MDA and H_2_O_2_. Feng et al. [10] stated that proper doses of selenium could reduce MDA accumulation in various plants. Selenium in plants can act as ROS at low concentrations, scavenging by increasing the antioxidant metabolism or act, enhancing the oxidative stress at high concentrations [28]. However, it can cause cell membrane degradation and senescence [21,30].

Both MDA and H_2_O_2_ are valuable biomarkers to detect oxidative stress in plants to monitor reactive oxygen species [27,28]. It was reported that H_2_O_2_ is a reactive molecule that takes part in reactions forming hydroxyl radical (OH), disrupting the physiological functioning of higher plants [28]. Jiang et al. [57] found that the MDA content in tobacco leaves significantly decreased at 6 mg kg^−1^ of selenium application concentration; however, 24 mg kg^−1^ selenium remarkably increased this parameter. Similarly, Cartes et al. [58] found that low-dose selenium treatments (≤6.0 mg kg^−1^) reduce the MDA content in ryegrass, and vice versa.

Plants under stress develop a network of enzymatic and non-enzymatic antioxidant systems to scavenge ROS and avoid oxidative damage. Selenium compounds can control the production and quenching of ROS damage, either directly or indirectly, through regulating antioxidants levels and activity [59]. In this study, a significant increase in SOD, CAT, POD, APX, and GR activation was observed by increasing selenium concentrations, which in agreement with the findings of Reference [51] in wheat, [34] in barley, in rice [27], in cowpea [28], and pepper fruits [60]. It was reported that the enhancement of enzymes activity is to lower the MDA and H_2_O_2_ concentration at levels lower than in the control plants, in which CAT converts H_2_O_2_ into H_2_O, and O_2_ [61] and GR catalyzes the reduction of oxidized glutathione into reduced glutathione (GSH), an essential compound in resistance to oxidative stress to decreased lipid peroxidation [62]. Silva et al. [28] and Reference [19] reported that SOD reduced ROS and transformation of H_2_O_2_ in H_2_O catalyzed by CAT and APX activity in plants treated with selenium. At the high doses of selenium used in the present experiment, selenium was observed to play a physiological role as a pro-oxidative agent, increasing the concentrations of H_2_O_2_, which may stimulate SOD activity. The same is true for APX, an enzyme that uses ascorbate as an electron donor to reduce H_2_O_2_ to H_2_O [63]. According to Stewart et al. [64], SOD, CAT, APX, and GR showed different activities at different selenium concentrations, showing that the element might stimulate or suppressing different enzymes depending on its concentrations.

## 4. Materials and Methods

### 4.1. Seeds and Sodium Selenite Solutions (Na_2_SeO_3_) Preparation Treatment

We carried the current experiment in the Laboratory for Advanced Substances Research Center (ASRC) at King Khalid University (KKU) in 2020. Seeds of quinoa (*Chenopodium quinoa* Willd) got from the Ministry of Agriculture in Abha. Healthy and viable seeds were selected, then sterilized with 5% sodium hypochlorite (NaOCl) solution (*v*/*v*) for 10 min, washed thoroughly with double-distilled water, and were soaked in sterile distilled water for one hour. Sodium selenite salt (Na_2_SeO_3_) with a molecular weight of 172.95 g M^−1^ was prepared with the following concentrations: 2.5, 5, 10, and 20 mg L^−1^ to be used compared to the control experiment solution (water only).

Quinoa seeds are planted in 15-cm perforated plastic pots from the bottom, filled with sand and peat moss (1:1) volume. Fifty seeds were planted in each pot approximately 1 cm deep and distanced equally. The pots placed in the greenhouse, and the temperature was set at 20–25 °C. All pots were irrigated three times a week with 200 mL of water for each pot for a week, and then the pots were divided into five groups, each group containing five replicates per treatment (250 seeds per treatment); treatments were 0, 2.5, 5, 10, and 20 mg L^−1^of Na_2_SeO_3_. The treatment continued for 23 days (30 days of starting planting), and at the end of the experiment, the plant samples of each pot were washed with water to remove the suspended soil in them for physiological tests.

The growth measurements were taken after 30 days of planting. Quinoa plants were harvested, and the measurements were averaged for ten plants from each pot, include shoot and root length (cm), number of leaves per plant, leaf area (cm^2^), fresh and dry weight (mg) of both roots and shoots (the samples were oven-dried at 70 °C to constant weight, and the dry weight was recorded. All growth measurements were taken at five replicates per treatment.

### 4.2. Determination of Relative Water Content

Three fully expanded young leaves from each treatment were taken. The fresh weight (FW) of each sample was recorded and immediately dipped in test tubes containing distilled water and kept in the dark. After 24 h, the leaves were taken out, wiped with the tissue paper, and the turgid weight (TW) was recorded. The samples were dried at 70 °C for 48 h, and the dry weight (DW) of each sample was determined. Relative water contents were calculated using the formula of Reference [65]. RWC = [(FW−DW)/(TW−DW)] × 100.

### 4.3. Determination of Photosynthetic Pigment Contents

For photosynthetic pigment determination, a three fresh leaf sample each of 0.2 g was extracted overnight with 80% acetone (5 mL), then grounded and homogenized. The homogenized mixtures were filtered, and the filtrates were raised to 25 mL by adding 80% acetone in each sample. The absorbance of the filtrates was determined at various wavelengths (663, 645, and 480 nm) using a UV-1900 BMS (Waltham, MA, USA) spectrophotometer. All readings were taken for three samples from each replicate per treatment. The chlorophylls (Chl.a and Chl.b), total chlorophyll (total chls), and carotenoids were calculated to determine their contents and expressed in mg/g FW following [66] formula.
Chl. a = [(12.7 × OD 663) − (2.69 × OD 645)] × V/1000 ΧW(1)
Chl. b = [(22.9 × OD 645) − (4.68 × OD 663)] × V/1000 ΧW(2)
Total chls = [(20.2 × OD 645) + (8.02 × OD 663)] × V/1000 ΧW(3)
Carotenoids = [(OD 480 + (0.114 × OD 663)] (0.638 × OD 645)(4)

### 4.4. Determination of Osmolytes Contents

Proline content was determined according to Reference [67]. Three fresh leaves with approximately 0.5g of each sample was homogenized in 5 mL of 3% sulfosalicylic acid. Ninhydrin reagent (2 mL) and glacial acetic acid 2ml were added to the test tube with 2 mL of extract. The mixture was boiled at 90 °C for thirty minutes. The reaction was ended in an ice bath. Then, 5 mL of toluene was added to the reaction mixture after cooling and vortex mixed for 15 s, kept in the dark for 20 min, at room temperature, to allow separation of the toluene layer from the aqueous solution. Each toluene layer was then carefully collected into a clean tube, and absorbance was read at 520 nm using a UV-1900 BMS (Germany) spectrophotometer. Free proline concentration was determined from a standard curve prepared using analytical grade proline and expressed as mg/g FW.

Total soluble sugars (TSS) were extracted and determined, according to Reference [68]. A 0.2 g sample from each of three fresh leaves was washed with 5 mL 70% (*v*/*v*) ethanol and homogenized in 5 mL of 96% (*v*/*v*) ethanol and then placed in a boiling water bath at 80 °C for 10 min. After cooling, the extract was centrifuged at 4000× *g* for 10 min, and the supernatant was stored at 4 °C for measurement. Total soluble sugar concentrations were determined by reacting 0.1 mL of the ethanolic extract with 3 mL of freshly prepared anthrone reagent (150 mg anthrone plus 100 mL of 72% (*v*/*v*) sulfuric acid) and placed in a boiling water bath at 80 °C for 15 min. After cooling, the mixture’s absorbance was recorded at 625 nm using a UV-1900 BMS (Waltham, MA USA) spectrophotometer to measure the quantity of total soluble sugars (mg/g FW) using a glucose standard curve.

### 4.5. Determination of Lipid Peroxidation (MDA) and Hydrogen Peroxide (H_2_O_2_) Content

Lipid peroxidation was analyzed in plant tissues through measurement of malondialdehyde (MDA) following [69]. Approximately 0.250 g fresh leaf sample (three leaves) was homogenized in 5 mL of 0.1% trichloroacetic acid (TCA) and centrifuged at 6000× *g* for 15 min. An aliquot of 1 mL was mixed with 4 mL of thiobarbituric acid (TBA), heated at 95 °C for 30 min, cooled in an ice bath, and then centrifuged. The supernatant was used to read the absorbance at 532 nm and 600 nm, respectively, using a UV-1900 BMS (Waltham, MA, USA) spectrophotometer. MDA content was calculated by subtracting a 532 nm reading output from a 600 nm reading output and using a (155 mM L^−1^ cm) absorption coefficient and expressed on a µM/g FW basis.

H_2_O_2_ was extracted by homogenized 50 mg of fresh leaves (three fresh leaves) with 3 mL of 50 mM of phosphate regulator (KH_2_PO_4_/K_2_HPO_4_, pH 6.5). A 3 mL of the extract was mixed with 1 mL of both 0.1% titanium sulfate and 20% H_2_SO_4_ and centrifuged at 6000× *g* for 15 min. The supernatant was then used to read the absorbance at 410 nm using a UV-1900 BMS (Germany) Please ensure the meaning has been retained. [70].

### 4.6. Determination of Antioxidant Enzyme Activity

Antioxidant enzymes, SOD, CAT, POD, APX, and GR, were determined spectrophotometrically. Approximately 0.5 g from each of three fresh leaf samples were grounded, and the homogenized mixture was filtered through a muslin cloth and centrifuged at 12,000× *g* for 10 min at 4 °C. Superoxide dismutase (SOD, EC, 1.15.1.1) activity was determined according to Reference [71] following the inhibition of photochemical reduction because of nitro blue tetrazolium (NBT). One unit of SOD activity was measured as the amount of enzyme required to cause 50% inhibition of the NBT reduction measured at 560 nm with a UV-1900 BMS (Waltham, MA USA) spectrophotometer. Catalase (CAT, EC, 1.11.1.6) activity was determined according to Reference [72]. The CAT activity was assayed by monitoring the absorbance decrease at 240 nm because of H_2_O_2_ disappearance (ɛ = 39.4 mM^−1^cm^−1^). Peroxidase (POD, EC, 1.11.1.7) activity was assayed according to Reference [73]. Variation because of guaiacol in absorbance was measured at 470 nm. Ascorbate peroxidase (APX, EC, 1.11.1.11) activity was assayed according to the method of Reference [74]. The oxidation of ascorbate was determined by the change in absorbance at 290 nm (ɛ = 2.8 mM^−1^cm^−1^). Glutathione reductase (GR, EC, 1.6.4.1) activity was measured after monitoring nicotinamide adenine dinucleotide phosphate (NADPH) oxidation for three absorbances was taken at 340 nm activity expressed as ∆ A340 min^−1^ mg^−1^ protein [75]. According to Bradford [76], total soluble protein was determined. About 0.5 g of fresh-ground leaves were well homogenized in phosphate buffer (0.05 M–pH 7.8) under cooling, filtered, and centrifuged for 10 min 12,000× *g* at 4 °C. Uv-Vis spectrum was measured at 595 nm.

### 4.7. Statistical Analysis

Data were subjected to analyzed one-way analysis of variance (ANOVA) and the honestly significant difference (HSD) at *p* < 0.05 probability level using Tukey’s test was used to compare the differences among treatment means using SPSS, V.22 for Windows [77] (SPSS, Chicago, IL, USA). The relative change (increasing or decreasing) in the effect of treatments was calculated referred to the control treatment.

## 5. Conclusions

In conclusion, selenium application at low concentrations induced physiological responses, such as plant growth, relative water content, photosynthetic pigments, proline, total soluble sugars, antioxidant enzymes, and promoted the antioxidant system. However, it is toxic at high concentrations, and the oxidative stress and antioxidant enzymes respond to high selenium levels. Thus, concentrations of selenium and its suitability for the plant type must be considered and not be stressful for the plant.

## Figures and Tables

**Table 1 plants-10-00719-t001:** Effect of various selenium levels (Na_2_SeO_3_) on plant growth parameters of plants’ growth after 30 days of selenium treatment. Number of samples 15:5 replications and 3 biological replicates.

Treatment Na_2_SeO_3_ mg/L	Plant Height (cm)	Root Length (cm)	Shoot Length (cm)	No. Leaves/Plant	Leaf Area (cm^2^)	Root Fresh Weight (mg)	Shoot Fresh Weight (mg)	Root Dry Weight (mg)	Shoot Dry Weight (mg)
0	35.75 ± 2.06	22.63 ± 1.99	29.42 ± 0.48	5.00 ± 1.00	4.32 ± 0.56	14.99 ± 1.73	22.23 ± 1.37	8.36 ± 0.35	13.31 ± 0.44
2.5	38.61 ± 2.95	26.40 ± 0.54	33.22 ± 0.68	7.33 ± 0.58	5.39 ± 0.52	17.36 ± 0.91	27.58 ± 1.65	11.04 ± 0.99	15.17 ± 0.56
5	43.29 ± 1.25	28.47 ± 0.51	37.55 ± 0.79	9.00 ± 1.00	6.70 ± 0.66	20.24 ± 1.32	31.31 ± 0.58	13.24 ± 0.65	19.05 ± 0.68
10	32.85 ± 1.61	19.55 ± 1.48	27.20 ± 0.49	4.67 ± 1.53	4.97 ± 0.93	12.87 ± 1.43	17.86 ± 1.45	6.96 ± 0.38	10.08 ± 0.20
20	27.82 ± 0.77	15.45 ± 0.70	21.91 ± 0.14	3.33 ± 0.58	2.64 ± 0.52	9.63 ± 0.40	13.13 ± 0.98	3.51 ± 0.47	8.99 ± 0.36
mean	35.66 ± 5.63	22.50 ± 4.94	29.86 ± 5.51	5.87 ± 2.26	4.80 ± 1.49	15.02 ± 3.91	22.44 ± 6.84	8.62 ± 3.51	13.32 ± 0.98
HSD * at *p* < 0.05	3.20	2.73	1.83	2.48	1.86	2.65	2.79	1.82	2.35

* HSD: an honestly significant difference at *p* < 0.05 probability level using Tukey’s test. Values in the table represent of means and standard deviations.

**Table 2 plants-10-00719-t002:** Effect of various selenium levels (Na_2_SeO_3_) on relative water content (RWC), chlorophyll-a. (Chl “a”, “b”), total chlorophyll (Chl “t”), and carotenoid content of quinoa plants after 30 days of selenium treatment. Number of samples 15:5 replications and 3 biological replicates.

Treatment Na_2_SeO_3_ mg/L	RWC	Chl “a”	Chl “b”	Chl “t”	Carotenoid
0	97.55 ± 0.34	4.87 ± 0.65	1.46 ± 0.45	5.92 ± 0.36	1.11 ± 0.07
2.5	98.21 ± 0.18	5.23 ± 0.52	2.38 ± 0.33	8.06 ± 0.10	1.56 ± 0.69
5	98.70 ± 0.43	6.12 ± 0.10	3.19 ± 0.18	8.63 ± 0.41	2.52 ± 0.34
10	95.64 ± 0.73	4.07 ± 0.23	1.02 ± 0.04	4.68 ± 0.37	0.98 ± 0.04
20	93.71 ± 0.61	3.15 ± 0.28	0.56 ± 0.09	3.63 ± 0.52	0.83 ± 0.09
mean	96.76 ± 1.96	4.69 ± 1.11	1.72 ± 1.01	6.18 ± 2.01	1.40 ± 0.70
HSD * at *p* < 0.05	0.75	0.62	0.42	0.58	0.55

* HSD: an honestly significant difference at *p* < 0.05 probability level using Tukey’s test. Values in the table represent of means and standard deviations.

**Table 3 plants-10-00719-t003:** Effect of various selenium levels (Na_2_SeO_3_) on proline, soluble sugars, malondialdehyde (MDA), and hydrogen peroxide (H_2_O_2_) content of quinoa plants after 30 days of selenium treatment. Number of samples 15:5 replications and 3 biological replicates.

Treatment Na_2_SeO_3_ mg/L	Proline mg/g FW	Total Soluble Sugars mg/g FW	MDA (µM/g FW)	H_2_O_2_ (µM/g FW)
0	12.61 ± 1.84	20.83 ± 0.85	24.41 ± 0.81	4.06 ± 0.72
2.5	15.55 ± 1.20	24.31 ± 0.69	19.10 ± 0.83	2.55 ± 0.20
5	18.58 ± 0.79	27.09 ± 0.91	20.68 ± 0.72	3.16 ± 0.09
10	22.29 ± 0.69	28.90 ± 0.51	27.25 ± 0.65	6.07 ± 0.75
20	25.55 ± 0.59	31.63 ± 1.61	29.37 ± 0.45	7.76 ± 0.41
mean	18.92 ± 4.87	26.55 ± 3.94	24.16 ± 4.04	4.72 ± 2.04
HSD * at *p* < 0.05	1.70	1.50	1.10	0.83

* HSD; An honestly significant difference at *p* < 0.05 probability level using Tukey’s test. Values in the table represent of means and standard deviations.

**Table 4 plants-10-00719-t004:** Effect of various selenium levels (Na_2_SeO_3_) on antioxidant enzyme activities of superoxide dismutase (SOD), catalase (CAT), peroxidase (POD), ascorbate peroxidase (APX), and glutathione reductase (GR) of quinoa plants after 30 days of selenium treatment. Number of samples 15:5 replications and 3 biological replicates.

Treatment Na_2_SeO_3_ mg/L	SOD (U/mg Protein)	CAT (U/mg Protein)	POD (U/mg Protein)	APX (U/mg Protein)	GR (U/mg Protein)
0	300.74 ± 2.04	250.99 ± 5.55	104.92 ± 1.70	112.89 ± 0.73	121.34 ± 0.71
2.5	306.70 ± 1.45	268.09 ± 3.84	109.10 ± 2.29	117.64 ± 1.88	126.86 ± 2.77
5	311.08 ± 1.83	269.33 ± 4.67	117.27 ± 0.33	124.51 ± 3.13	132.56 ± 1.30
10	320.45 ± 1.07	280.71 ± 4.93	122.34 ± 1.28	131.74 ± 1.30	139.50 ± 2.87
20	327.87 ± 2.31	284.12 ± 3.34	126.59 ± 0.678	138.02 ± 1.91	149.49 ± 4.13
mean	313.37 ± 10.14	270.65 ± 12.64	116.05 ± 8.42	124.96 ± 9.59	133.95 ± 10.42
HSD * at *p* < 0.05	2.80	6.85	2.25	2.95	4.00

* HSD: an honestly significant difference at *p* < 0.05 probability level using Tukey’s test. Values in the table represent of means and standard deviations.

## Data Availability

The data presented in this study are available on request from the corresponding author.

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
