# Peer review of "Antioxidant Enzymatic Activities and Growth Response of Quinoa (*Chenopodium quinoa* Willd) to Exogenous Selenium Application"

_plants, 2021, doi:10.3390/plants10040719_

Round 1
Reviewer 1 Report
This manuscript is about an interesting concept, but it has a number of major shortcomings that need to be addressed.
Materials and methods:
- Paragraph 4.2: What do you mean by the term "flag leaves"? Do you refer to a specific phenological growth stage of quinoa according to the BBCH scale?
- Paragraphs 4.3-4.6: Please describe in details which plant tissues were sampled? Which leaves were collected? And why have you chosen those leaves.
Results:
- The legends of tables and figures need to be rewritten, in order to be self-explanatory and to provide sufficient information without referring to the related text in the manuscript. For example, you say in your legends "after 30 days": 30 days after what?
- The figures should be redrawn, in order to effectively present your data. For example, we do not need mean values in the x-axis but we do need error bars.
- The samplings concerning data presented in Table 1 are not described in the Materials and methods.
Unfortunately, I cannot proceed to the evaluation of the results of this study unless those shortcomings will successfully be addressed.
Author Response
Dear Professor
Thank you very much for your valuable notes and suggestions you raise to improve our work. please find attached file regarding our responses.
Best regards

Reviewer 2 Report
The manuscript "Antioxidant enzymatic activities and growth response of Qui-2 noa (Chenopodium quinoa Willd) to exogenous selenium appli-3 cation" Posted in plants is interesting, unfortunately requires a lot of corrections to be accepted. the text must be subject to linguistic proofreading.
Please correct the following errors:
line 19 - remove the term dramatically throughout the text and replace it eg significantly
throughout the text, selenium is capitalized incorrectly, as is sulfur (e.g. lines 39,44)
- there is chaos in the use of the alternating abbreviations Se or Selenium
Introduction
This section should be supplemented with information about the sources of the increased amount of selenium in the environment, what it results from
- Latin names of plants and families are written in italics (lines 56, 57, 64)
- bad records of chemical formulas, e.g. H2O2; selenium compounds in text and graphs
- Lowercase quinoa should be in the text
- headers should be written without dots and a colon
- Figure 2 - "t" explain abbreviations in figure description
line 134 - lowercase glutathione
- explain the abbreviation for the description of Figure 5
Discussion:
- line 149 - add the name before citing
- line 163 - Chlorophyll from lowercase and further in the text
- line 165 - give citation at the end of the sentence
- line 253 - remove (Silva et al.)
- lines 267,354 - remove the abbreviation G.R. which we write without dots !!
- line 321 - lower case toluene
- lines 346,351,369, - correct units
Conclusions
Correct the whole thing, especially the sentences from lines 377-381
References
also have errors in dates, please check and correct
Author Response
Dear Professor
Thank you very much for your valuable comments and suggestion you offer to improve our work. Please kindly find an attach file with response to all comments you suggested.
Best regards

Reviewer 3 Report
The manuscript submitted for review addresses interesting issues in the biology of plant stress. It is merotoric enough to consider publishing it in Plants, but in my opinion a major and extensive revision of this manuscript is necessary before that can happen.
1. The first case concerns the formal page of the manuscript. While working on the new version of the text, please approach the chemical side of this work with great care and attention to detail. Throughout the manuscript, the incorrect writing of chemical formulas is very striking. Number of atoms in chemical compound please exppress as subscript.
2. The second issue concerns the graphic presentation of the results. I am asking the authors to prepare completely new charts. The present ones are not legible. Please prepare simple bar graphs without adding a 3D effect. Prepare new graphs in a more careful manner than can be seen in the present form.
3. The third issue is also related to the presentation of the results on charts. The authors do not present appropriate error bars, showing, for example, standard deviation or other estimator of statistical error. When you prepare a new chart, please take this into account. Without such data, the presented data is not entirely reliable.
Therefore, I am asking Editors to reject this manuscript and to allow the authors to re-submit the new manuscript but as a new submission.
Author Response
Dear Professor
Thank you very much for your kind suggestions and comments which helped in improving our work.
Kindly find attached the responses to your valuable comments,
Best regards

Round 2
Reviewer 1 Report
I would like to thank the authors for their responses to my comments. Unfortunately, I feel that there still are a number of shortcomings that need to be addressed.
We need error bars in the figures. And we need to know if those error bars represent SD or SE (also for the supplementary material). We also want to know what type of error is represented in Table 1.
Moreover, you mention at paragraph 4.1 that five replicates per treatment were taken for growth measurements. How many replicates were taken for every one of the determinations described in paragraphs 4.2-4.6?
You need to redrawn your figures, especially in terms of the x- and y- axis labels. You may take a look at other published papers or consult an expert in order to prepare figures that will efficiently highlight your data.
Author Response
Dear respective professor
Thank you very much for the suggestions and valuable comments you raised that improve our work.
Please find attached the file of our response. All corrections are inserted in the main manuscript.
Best regards

Reviewer 2 Report
The manuscript still has editing errors, especially in spelling with capital letters, which should be corrected e.g.
Lines 188 -Sodium
212 - Selenium
305 Sodium ............. Preparation ect
Author Response
Dear respective professor
Thank you very much. We did all correction and corrected in the main manuscript. Thank you again for the valuable comments you raise that improve our work
Best regards
Reviewer 3 Report
The revised manuscript was submitted for re-evaluation. However, I find that it still needs improvement. I still do not see error bars in the figures. Without them, a substantive evaluation of these results is impossible. Please correct this in the next round of revision. Still some figures contain formulas of chemical compounds incorrectly written. It should also be ensured that in the file that is presented for evaluation, the figures are "anchored" in the right place so that the appearance of the manuscript is aesthetic.
Author Response
Dear respective professor
Thank you very much for the valuable comments and suggestions raised that improved our work. We are pleased to attach our response. All corrections are inserted in the main manuscript in different colors.
Best regards

Round 3
Reviewer 1 Report
I would like to thank the authors for their responses. I think that one last thing is missing in the materials and methods section: a description and a reference of the method used for total protein determination, since the enzymes activities are expressed as Units/mg Protein.
Author Response
Dear professor
Thank you very much for your valuable comment. We are sorry for not include the total protein determination. We have used Bradford assay for protein determination. We inserted the method and reference in the materials and method section Best regards
Reviewer 3 Report
Thank you very much for responding to my comments. I believe that the authors responded very well to my comments and significantly improved the manuscript. This manuscript can be accepted for printing in its present form.
I send greetings.
Author Response
Dear professor
Thank you very much for your efforts and help during reviewing our manuscript which made our work have more value.
Best regards